# A Study of Modern Eco-Friendly Composite (Geopolymer) Based on Blast Furnace Slag Compared to Conventional Concrete Using the Life Cycle Assessment Approach

Alireza Esparham [1],*, Nikolai Ivanovich Vatin [2],*, Makhmud Kharun [3] and Mohammad Hematibahar [2]

1   Department of Environmental Engineering, University of Tehran, Karaj 999067, Iran
2   Institute of Civil Engineering, Head of Research Quarter of Peter the Great St. Petersburg Polytechnic University, 195251 St. Petersburg, Russia
3   Department of Reinforced Concrete and Stone Structures, Moscow State University of Civil Engineering, 129337 Moscow, Russia
*   Correspondence: alireza.esparham@ut.ac.ir (A.E.); vatin@mail.ru (N.I.V.)

**Abstract:** By posing the question of what will be the definition of sustainable development in the future, it can almost be seen that the principle of "no waste" and the production of new materials with less of a negative environmental impact will have a high priority. To further develop environmentally friendly materials, it is necessary to know about the environmental drivers of new materials as well as to evaluate the environmental effects of conventional materials in construction. According to the definitions of sustainable development and sustainable materials, materials with characteristics such as having low energy consumption, sufficient durability, good physical and chemical properties, while simultaneously reducing pollution should be used. Geopolymer materials may be a reasonable option. In this research, two production processes based on blast furnace slag and ordinary concrete (Portland cement) for one cubic meter of geopolymer concrete have been investigated. To investigate, inputs (materials and energy) and outputs (relevant environmental pollutants) of both systems were determined and a life cycle assessment (LCA) was measured using the Center of Environmental Science of Leiden University (CML) and cumulative exergy demand (CED) quantification methods of SimaPro V.9 software. The results showed that the production system of one cubic meter of conventional concrete has maximum environmental effects in all classes except in the destruction of the ozone layer, and the system of producing one cubic meter of geopolymer concrete based on slag has much less environmental effects than the normal concrete system. It also consumes 62% less directly during its lifetime. As a result, geopolymer concrete may be a suitable alternative to traditional concrete as a sustainable material.

**Keywords:** geopolymer; sustainable development; life cycle assessment; greenhouse gases; energy consumption



## 1. Introduction

Many industries have used the word "sustainability". Sustainable development has different definitions, but one of the most common ones is that today's generation should not threaten the ability of future generations to meet their own needs [1]. The four pillars of sustainable development are: economic, environmental and legal support and social development [2]. On the one hand, the manifestation of sustainable development in a built environment is called sustainable architecture. On the other hand, separating this issue from other economic, cultural and social factors is challenging. According to Richard Rogers, the goal of sustainable design is to meet future needs without depleting remaining natural resources for future generations. Sustainable design in buildings is related to resource efficiency, low energy consumption, flexibility and longevity. According to Jung Jin Kim,

the three principles of resource conservation, life-cycle-based design, and human-based design comprise sustainable architecture [3].

Today, one of the most important issues in the construction of buildings and settlements is the use of sustainable architecture. The environmental effects and pollution effects must be determined in sustainable urban architecture. Industrial and production activities should be coordinated with environmental concerns to reduce pollution

Natural resource restrictions and human priorities have been proven to influence the lands are utilized to support humans. Cities now consume three-quarters of the world's energy and then contribute to 75 percent of the global pollution [4].

Furthermore, the United Nations estimates that by 2050, 68% of the world's population will live in cities [5]. As a result, in the discussion of sustainable development, attention should be paid to the increasing dominance of cities and its direct and indirect consequences. Large cities act as hubs for large networks of critical infrastructure services. Consequently, the adaptability and robustness of urban infrastructure is critical for long-term development [6].

The characteristics of sustainable materials are mentioned in the discussion of sustainable architecture, including energy consumption and the amount of pollution produced, as well as that of having sufficient durability and strength [7–9].

Materials from recycled or harmless end-of-use waste are also included in sustainable building materials, as is the focus on on-site waste management. This means that most of the waste generated in the city or a larger area (industrial suburb) can be recycled and put to a number of uses, including construction. The quality and price of these materials are two important factors that must be considered. Sustainable materials are renewable materials that have a good effect on employment and contribute to economic activities based on economy, environment and energy [10–12].

Moreover, the promising positive-sustainability properties of geopolymer concrete, as well as the use of by-products and the increased need for natural resources, will encourage its increased adoption by the concrete industry and in other parts of building materials compared to conventional concrete with low embodied carbon. However, to justify the improved sustainability of geopolymer concrete compared to conventional concrete, a comprehensive life cycle assessment that considers both upstream and downstream impacts is required [13].

On the other hand, the manufacture of Portland cement and its use in building materials, such as concrete, releases significant amounts of carbon dioxide into the atmosphere (one ton of Portland cement produces nearly one ton of carbon dioxide). On the other hand, it has become one of the most important environmental concerns in the world. Carbon dioxide (65%) is a key contributor to global warming, and the Portland cement manufacturing process emits 5–7% of total global carbon dioxide emissions [14–16].

Geopolymers harden quickly and have a high initial strength, and after 28 days, the ultimate compressive strength reaches 100 MPa. Geopolymers have a permeability of 9–10 cm/s, low alkaline expansion, low shrinkage and high resistance to acids, sulfates, corrosion and fluidity properties similar to Portland cement [17–20]. The geopolymer concrete has impressive mechanical properties as well as chemical properties inspired by conventional concrete [21,22]. For example, Kuri et al. [23] studied the effect of waste glass as coarse aggregates on conventional concrete and geopolymer concrete. Their results show that the tensile strength of geopolymer concrete was more than 5 MPa, while the conventional concrete tensile strength was more than 3 MPa. In addition, the compressive strength result of conventional concrete was more than 50 MPa, while geopolymer concrete had more than 60 MPa of compressive strength. Overall, any source of silica and alumina that can be dissolved in an alkaline solution can serve as a geopolymer and polycondensation precursor [24–26].

Mines, power plants, urban, agricultural, and construction wastes, as well as a variety of aluminosilicate sources that are produced in large numbers in every country today, can all be used to make geopolymer materials, concrete and building. There is also the use of

fire and insulation. Some of these waste materials (such as fly ash and slag) are currently used exclusively as pozzolans in the manufacture of Portland cement. Granulated blast furnace slag is one of the most common sources of geopolymerization. Slag is a by-product of iron furnaces, which can be one of the best primary sources of geopolymerization due to its amorphous structural character [27–30].

When compared to cost-effective systems, less is known about the environmental consequences of geopolymer production. The technical literature can provide some estimates or approximations of unit environmental characteristics (without specific system constraints) from a quantitative point of view. Most of them discuss carbon dioxide emissions and compare geopolymer (concrete) production to Portland cement (concrete) production [20,24,27–30]. A life cycle assessment is the most common method for estimating environmental impacts. LCA enables a comprehensive analysis of products and services. This research analyzes the possibilities of making geopolymer concrete using blast furnace slag (as a source of aluminosilicate) for sustainable urban development, reducing greenhouse gases, reducing energy consumption and comparing it with traditional concrete (Portland cement) using LCA.

## 2. Materials and Methods

Life cycle assessment (LCA) is a technique for evaluating all inputs and outputs of a product (inputs and outputs), process or service (life cycle inventory), waste assessment, human health effects and ecological effects (impact assessment). Moreover, the interpretation of the evaluation results (interpretation of the life cycle) in the entire life cycle of the product or process is investigated. ISO standards in the life cycle environmental impact assessment section have four processes.

### 2.1. ISO Standards in the Life Cycle Environmental Impact Assessment

2.1.1. Classification

At this step, the documented and quantified environmental interventions in the inventory analysis are assigned to the predetermined effect classes on a strictly quantitative basis.

2.1.2. Characterization

At the step of assessing the effect of evaluating the impact of environmental interventions given to a class, the specific effect is quantified based on the customary unit for the same class, and the sum is reported as a score (index result).

2.1.3. Normalization

Normalization of a product's life cycle evaluation data is defined by ISO 14,042 as "calculating the range of index results relative to reference information." Reference information may refer to the specified community (e.g., Europe or the world), the individual (e.g., a citizen) or other systems within the given time period. The primary objective of normalization is to improve the comprehension of the significance and relative distribution of study outcomes for each product system. The result of the normalization phase indicates which effect category (such as global warming or acidification) creates the greatest environmental burden during the product's life cycle [31].

2.1.4. Weighting

Normalization of a product's life cycle evaluation data is defined by ISO 14,042 as calculating the index results relative to reference information range [31]. Reference information may notify that the specified community (e.g., Europe or the world), the individual (e.g., a citizen). The primary objective of normalization is to improve the comprehension of the significance and relative distribution of study outcomes for each product system. The result of the normalization phase indicates which effect category (such as global warming or acidification) creates the greatest environmental burden during the product's life cycle.

This section may be divided by subheadings. It should provide a concise and precise description of the experimental results, their interpretation, as well as the experimental conclusions that can be drawn [30,32,33].

### 2.2. Defining the Goals and Boundaries of the System

This study compares geopolymer concrete with conventional concrete to see which has the least environmental impact and uses the least resources and energy.

The operational (functional) unit of one cubic meter of concrete was chosen for comparison in this research in order to compare two products or two types of production methods of geopolymer concrete and conventional concrete.

Considering the time limit of the research and the lack of information needed to conduct extensive research, it was decided to determine the boundaries of the system according to Figures 1 and 2. Because a life cycle assessment is both extensive, complex and detailed approach—a life cycle in terms of input and output —the boundaries of the system must be determined with high precision. In a comparative LCA, life cycle stages that are comparable for both substances may be omitted. Consequently, the study will focus on the production stage as well as the supply networks associated with resource extraction (Figure 1) [34].

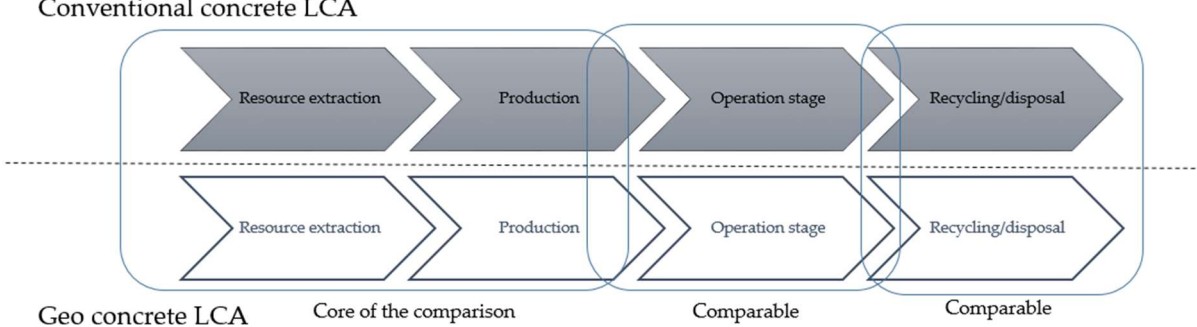

**Figure 1.** System boundaries for comparing the production of geopolymer concrete and cement concrete.

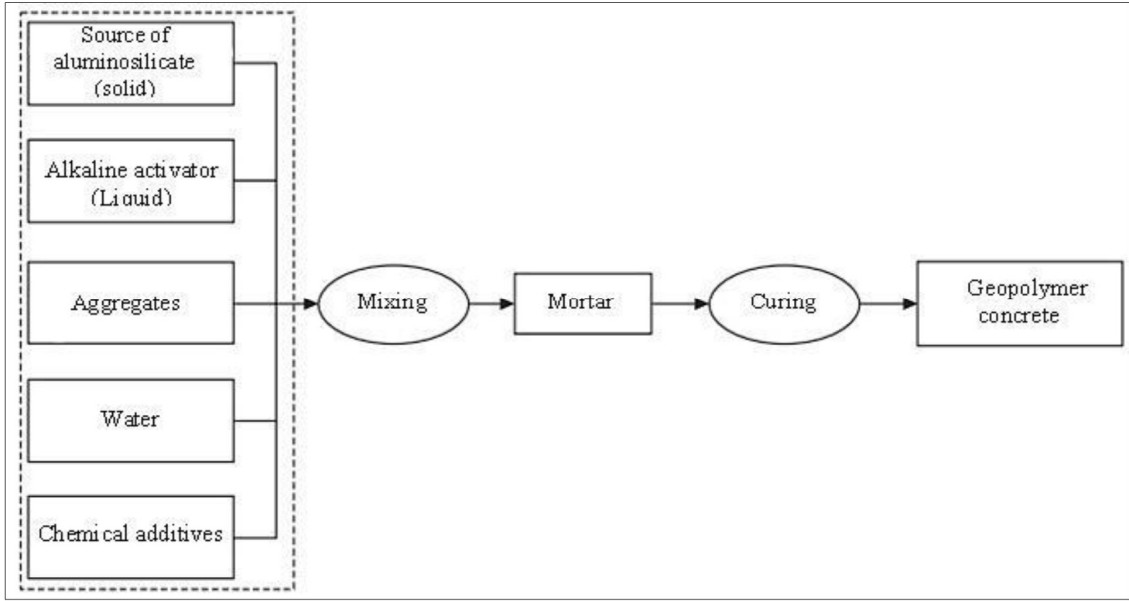

**Figure 2.** System boundaries for comparing geopolymer compositions.

Although the transportation of raw materials contributes to the environmental profile, most life cycle assessment papers do not include system boundaries because the efforts to transport the required raw materials vary widely around the world. However, in this article, an attempt has been made to evaluate the impact of transportation processes in a specific area (a construction workshop located 45 km from an Isfahan iron smelter in Isfahan Metropolis).

### 2.3. Prepare a Life Cycle List

Life cycle inventory (LCIs) were performed on data from multiple sources in this study. Equinont 12 databases in SimaPro provided primary data for raw materials, auxiliaries and semi-finished products.

It is one of the most comprehensive databases created by the Swiss Life Cycle Inventory Center in partnership with the Federal Institute of Technology in Zurich. This database contains 10,000 public processes in energy, transportation, building, chemical, agricultural, cardboard, laundry and waste management [35].

Materials and procedures were chosen from the SimaPro software database based on information regarding the availability of concrete raw materials and the opinions of industry experts. Granulated blast furnace slag, or GGBS, is the primary source of geopolymerization discussed in this study. The XRF analysis of the bought slag from Isfahan Steel Company is provided in Table 1. Table 2 also offers information on the mixing plan for geopolymer concrete and conventional concrete, which results in the creation of concrete with a 28 day compressive strength of approximately 80 MPa [29,36].

**Table 1.** XRF of blast slag.

| Cl | MnO | $Na_2O$ | $K_2O$ | MgO | $Fe_2O_3$ | CaO | $Al_2O_3$ | $SiO_2$ |
|------|------|------|------|------|------|------|------|------|
| 34.4 | 11.2 | 37 | 0.6 | 9.8 | 0.68 | 0.6 | 1.58 | 0.002 |

**Table 2.** Differences between geo concrete and conventional concrete.

| Material | Geo Polymer Concrete ($kg/m^3$) | Conventional Concrete ($kg/m^3$) |
|------|------|------|
| OPC | - | 600 |
| Blast furnace slag | 410 | - |
| Sodium Silicate solution | 120 | - |
| NaOH (50%) | 80 | - |
| Superplasticizers | 8 | 25 |
| Water | 31 | 180 |
| Gravel | 850 | 1020 |
| Sand | 850 | 524 |
| Total | 2349 | 2349 |

The time it takes to apply the concrete welter mix and the power consumed by the concrete welter mixer were used to calculate the power consumption in concrete construction (power consumption of concrete production 0.75 kwh). In this study, a carbon dioxide emission factor of 0.0033 $kg/m^3$ was determined based on Turner et al. [37] investigations on the process of creating carbon dioxide concrete, and a carbon dioxide emission factor of 0.009 $kg/m^3$ was computed for final payment and, eventually, concrete processing [37].

### 2.4. Impact Assessment

The possible environmental consequences of the environmental inputs and outputs shown in the LCI are evaluated through an impact assessment. LCI has been interpreted as potential environmental consequences using various models in environmental systems (such as global warming due to greenhouse gas emissions) [38]. Depending on the product

or system, several methods and tools are used to assess environmental consequences in LCA. SimaPro is one of the most practical and complete of these programs. Simapro is a professional tool for studying the environmental characteristics of a product or service. This software achieves this in a methodical and consistent way and provides the possibility of finding the best project solutions. SimaPro comes with a number of impact assessment methods to calculate environmental impact assessment results. The classification of effects, environmental models and identification factors differ between these methods.

In this study, the practical and comprehensive method of CML in quantifying the assessment of environmental effects (such as global warming potential or greenhouse gas effect) and the CED method for evaluating the life cycle energy resources of conventional concrete and geopolymer were investigated in SimaPro V.9 software.

### 2.4.1. CML Method (Center of Environmental Science of Leiden University)

Each material's life cycle must be considered to understand its environmental effects [7,39,40]. In this cycle, energy consumption and pollutants are vital for sustainable design [40]. In this work, the widely utilized CML approach was employed to analyze material environmental degradation. A team of CML-led scientists presented impact categories and descriptive methodologies in 2001.

CML is a procedure used to estimate the measurement of the environmental impact caused by the product. This method uses various impact categories such as eutrophication, ionization radiation, aquatic ecotoxicity, land use, and human toxicity. In fact, the CML method is an impact assessment approach that limits quantitative modeling to early stages in the chain of cause and effect. Results are grouped into midpoint groups based on common mechanisms (e.g., climate change) or broadly agreed upon groups including eco-toxicity [19].

In SimaPro software, there are two variants of this CML method: one with 10 impact categories, and another with further adjustments to the categories for different time periods [41].

### 2.4.2. CED Method (Cumulative Exergy Demand)

The cement industry sector and the production of materials, such as concrete, account for a significant amount of energy consumption in the world, and the depletion of fossil resources and global warming are both directly affected by energy consumption. Cement used in ordinary concrete is one of the most important industries. It consumes energy, especially fossil resources. Therefore, energy production with a large amount of fossil fuel has been developed and the emission of polluting gases in the environment has increased [40,42]. On the other hand, sustainable architecture seeks efficiency in energy consumption in the use of building materials [12]. Therefore, the CED method is chosen to evaluate energy resources in this research. CED represents the direct and indirect energy used during the life of a product, including the energy consumed during the extraction, production and disposal of raw and auxiliary materials [43,44].

The appropriateness of the fossil cumulative energy demand (CED) as an indicator for the environmental performance of products and processes is explored with a regression analysis between the environmental life-cycle impacts and fossil CEDs of 1218 products divided into the product categories "energy production", "material production", "transport" and "waste treatment" [45].

Exergy is the most functional work of the production function: Exergy is the potential of the system to bring about change and the energy available for use in balancing that system with the environment. The CED Index is defined as the sum of all the resources required to supply a process or product. CED is defined as the MJ equation. Moreover, it is divided into eight subgroups: fossil fuels, nuclear, hydroelectricity, biomass, other renewables, water, minerals and metals. This index is determined by the total number of allergens removed from nature to create products or services in desired systems [46].

### 3. Results and Discussion

The cement industry and the manufacture of materials such as concrete consume a significant amount of energy globally, and energy consumption has a direct impact on the reduction of fossil resources and global warming. One of the most important roles is the production of cement, which is used in ordinary concrete. As a result, the production of energy from fossil fuels has increased, as well as the emission of toxic gases into the atmosphere. On the other hand, sustainable design pursues energy efficiency through the use of building materials. As a result, in this study, the CED approach was adopted to evaluate energy resources. CED is the total amount of direct and indirect energy consumed by a product during its lifetime, including the energy consumed in the extraction, manufacture and disposal of raw materials and auxiliary materials. In this section, CML and CED are used to evaluate the output of SimaPro software and the findings of the study. First, graphs related to each system are detailed based on their relative contribution to each effect category, followed by a table detailing the results of each system. Finally, a comparison has been made between two types of geopolymeric concrete systems and conventional concrete.

### 3.1. Geopolymer Concrete System by CML Method

As depicted in Figure 3, the results of SimaPro software in the CML method depict the impact assessment in 10 effect classes and provide the necessary information, such as the aluminosilicate source of the blast furnace slag used, sodium silicate and sodium hydroxide in the concrete, as well as the amount of electricity consumed on-site in cubic meters of concrete. All information entered in the software regarding the impact in each impact category are included. The effect of resource depletion (fossil fuels) reveals the energy consumption of a system. In other words, it indicates the extent to which a system can be utilized to reduce fuel consumption. The potential for the acidification of a pollutant is determined by its capacity to produce positive hydrogen ions in the acidification impact class.

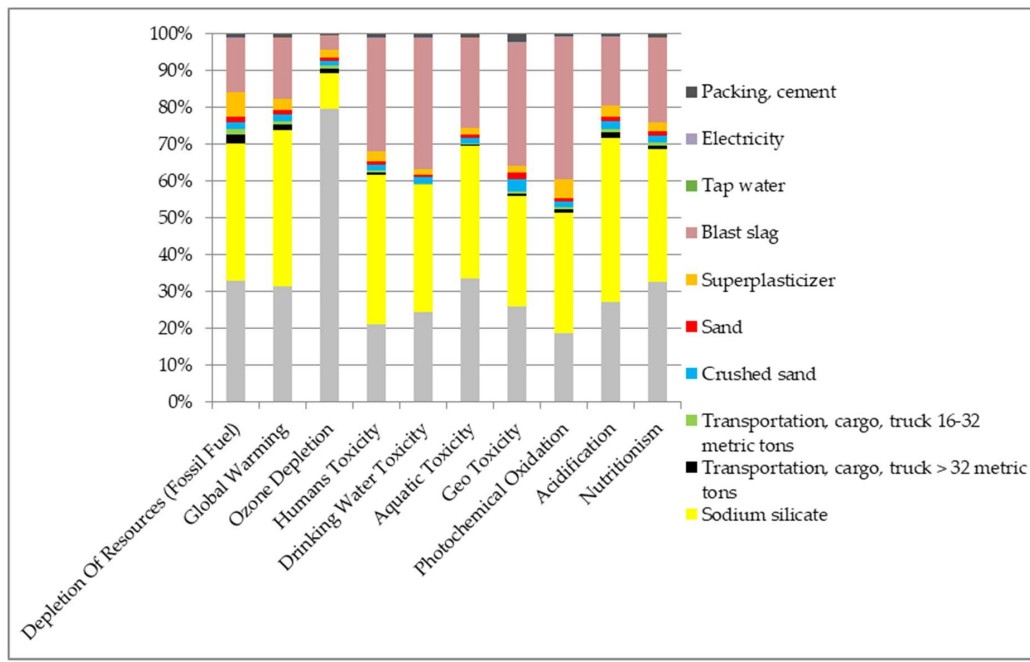

**Figure 3.** The graph of the results of the life cycle evaluation of the production process of one cubic meter of geopolymer concrete by CML method.

In the nutritional impact class, nitrogen and phosphorus were found to affect algal bloom. In the category of global warming effect, climate change associated with the generation of carbon dioxide greenhouse gases has been researched. The measurement standard

for the ozone depletion class is stratospheric ozone per chlorofluorocarbon generated. In the category of human toxicity potential, the threat to human health has been evaluated. In the category of photochemical oxidation effects, chemicals that contribute to the generation of photochemical ozone are quantified in terms of ethylene. Other classes also explain the toxicity of land, drinking water and marine environments. The geopolymer is more harmless than conventional concrete for the ozone layer. For example, Imtiaz et al. [47] proved that the geopolymer concrete, was more harmless for the ozone layer.

The results showed that the source of alumina silicate used in the geopolymer concrete system with a share of 38.7% has the greatest effect of environmental degradation in the photochemical oxidation effect category and the least effect of environmental degradation in the biological (fossil) reduction fuel classification and in global warming, at less than 17%. It is also noteworthy that consumable alumina silicate sources have the least impact on the ozone layer and constitute only 4.02% of the total.

The electricity of the geopolymer concrete system has an effect of 0.4% on the destruction of the environment. Sodium silicate has the greatest effect of acidification (44.5%) and the least effect of destroying the ozone layer (9.68%). Sodium hydroxide used in geopolymeric concrete systems has the greatest effect of destroying the ozone layer (79.6%) and the least effect of photochemical oxidation (18.6%).

The superplasticizer used in the geopolymer concrete mixing plan has the effect of reducing resources (fossil fuel) by 6.52% and less than 5% in other classes.

Environmental deterioration has a relatively small impact of less than 2.5 percent in virtually all classifications, including consumable sand and the packaging and transportation of consumables (Table 3, Figure 3).

### 3.2. Conventional Concrete System by CML Method

According to Figure 4, the Portland cement used in the concrete system has the highest environmental degradation effect in the global warming effect class with a share of 84.4%, and then it has a contribution of 71.4% in the earth toxicity effect class as well as the lowest effect in the human toxicity effect class, with a 26.6% share. The sand utilized has the largest influence on resource depletion with 27.8% and less than 3% in other impact classes. Transportation, sand use and cement packaging all contribute less than 2% to environmental degradation (Table 4).

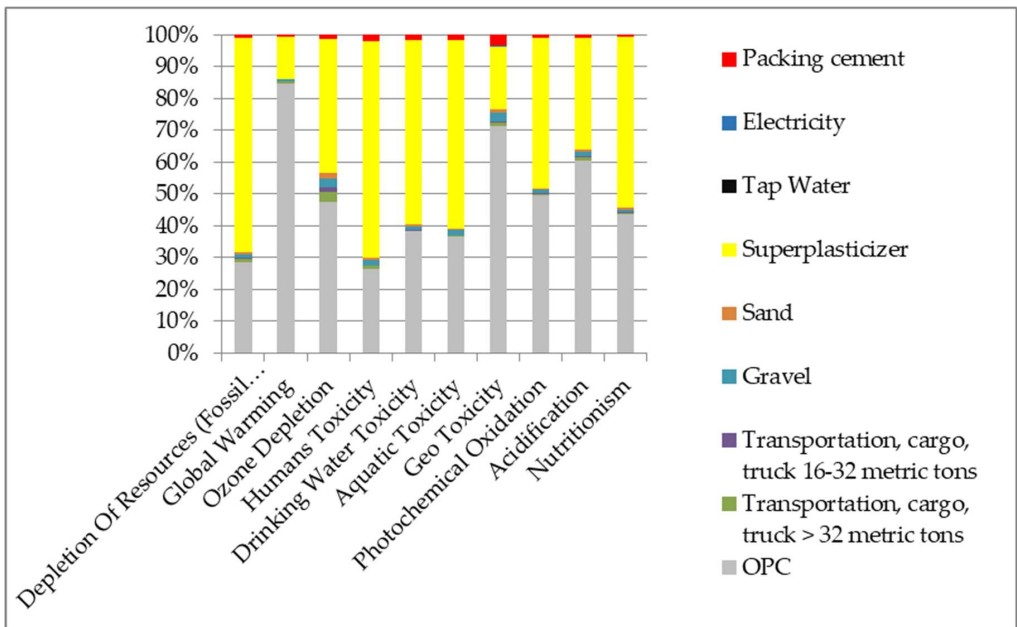

**Figure 4.** Diagram of the life cycle evaluation results of the production process of one cubic meter of concrete made of ordinary Portland cement by CML method.

**Table 3.** The values obtained from the life cycle evaluation of the production process of one cubic meter of geopolymeric concrete by CML method.

| | Scale | Total | Sodium Hydroxide | Sodium Silicate | Transportation, Cargo, Truck >32 Metric Tons | Transportation, Cargo, Truck 16–32 Metric Tons | Crushed Sand | Sand | Superplasticizer | Blast Slag | Tap Water | Electricity, High Voltage | Packing, Cement |
|---|---|---|---|---|---|---|---|---|---|---|---|---|---|
| Depletion Of Resources (Fossil Fuel) | MJ | 1780.29 | 1780.29 | 587.2391242 | 665.9346869 | 42.76416461 | 23.10947838 | 36.68141802 | 25.08641368 | 116.108373 | 259.3437742 | 0.004303545 | 6.558797671 |
| Global Warming | kg CO2 eq | 172.37 | 172.37 | 54.3663739 | 73.05409991 | 2.719436803 | 1.524176971 | 3.202615852 | 1.973379799 | 5.102638372 | 28.57330153 | 0.000431339 | 0.480197931 |
| Ozone Depletion | kg CFC-11 eq | $4.54503 \times 10^{-13}$ | $4.54503 \times 10^{-13}$ | $3.22916 \times 10^{-5}$ | $3.9997 \times 10^{-6}$ | $5.22418 \times 10^{-7}$ | $2.60451 \times -10^{-7}$ | $5.11552 \times 10^{-7}$ | $3.75716 \times -10^{-7}$ | $8.32716 \times 10^{-7}$ | $1.63083 \times 10^{-6}$ | $1.90434 \times 10^{-10}$ | $4.4913 \times 10^{-8}$ |
| Humans Toxicity | kg 1,4-DB eq | 150.26 | 150.26 | 31.99070253 | 60.70995363 | 1.080715082 | 0.651857305 | 2.654701489 | 1.252863178 | 4.098172313 | 46.12563399 | 0.000340619 | 0.112641226 |
| Drinking Water Toxicity | kg 1,4-DB eq | 100.01 | 100.01 | 24.51684252 | 34.42459116 | 0.236766799 | 0.281692705 | 1.566163635 | 0.817490035 | 1.537404913 | 35.53233986 | 0.000415486 | 0.09112208 |
| Aquatic Toxicity | kg 1,4-DB eq | 268,516.1 | 268,516.1 | 89,853.73417 | 97,477.27011 | 739.6282223 | 887.6853829 | 3891.182437 | 2197.636085 | 5172.491433 | 65,587.40947 | 0.862091126 | 289.8838903 |
| Geo Toxicity | kg 1,4-DB eq | 0.46 | 0.46 | 0.120164362 | 0.136888679 | 0.004060262 | 0.002267745 | 0.015750411 | 0.007428518 | 0.008644984 | 0.154197251 | $4.09601 \times 10^{-6}$ | 0.000233202 |
| Photochemical Oxidation | kg C2H4 eq | 0.06 | 0.06 | 0.010410215 | 0.018365984 | 0.000487568 | 0.000296186 | 0.000883413 | 0.00046984 | 0.002904168 | 0.021634804 | $6.66295 \times 10^{-8}$ | $7.10424 \times 10^{-5}$ |
| Acidification | kg SO2 eq | 0.93 | 0.93 | 0.252976627 | 0.413207868 | 0.01406108 | 0.007850201 | 0.01960925 | 0.012814056 | 0.027893246 | 0.172695984 | $1.91035 \times 10^{-6}$ | 0.00100851 |
| Nutritionism | kg PO4—eq | 0.36 | 0.36 | 0.116161497 | 0.129086757 | 0.003336236 | 0.002682322 | 0.007321977 | 0.004455431 | 0.007716382 | 0.082380229 | $1.8924 \times -10^{-6}$ | 0.000584822 |

**Table 4.** The values obtained from the life cycle evaluation of the production process of one cubic meter of concrete made of ordinary Portland cement using the CML method.

| | Scale | Total | POC | Transportation, Cargo, Truck >32 Metric Tons | Transportation, Cargo, Truck 16–32 Metric Tons | Gravel | Sand | Superplasticizer | Tap Water | Electricity, High Voltage | Packing, Cement |
|---|---|---|---|---|---|---|---|---|---|---|---|
| Depletion Of Resources (Fossil Fuel) | MJ | 4662.7 | 1324.2089 | 57.38895249 | 21.46837686 | 44.2766292 | 23.90587679 | 3148.6114 | 0.043035446 | 5.730002647 | 37.1085487 |
| Global Warming | kg CO2 eq | 742.88 | 629.9156266 | 3.620537051 | 1.41667757 | 3.865745724 | 1.880514878 | 98.87939972 | 0.004313395 | 0.382606396 | 2.911557855 |
| Ozone Depletion | kg CFC-11 eq | $2.33857 \times -10^{-13}$ | $9.90596 \times -10^{-6}$ | $7.01847 \times 10^{-7}$ | $2.62883 \times -10^{-7}$ | $6.17474 \times -10^{-7}$ | $3.58035 \times -10^{-7}$ | $8.75444 \times -10^{-6}$ | $1.90434 \times -10^{-9}$ | $4.28296 \times -10^{-8}$ | $2.34744 \times -10^{-7}$ |
| Humans Toxicity | kg 1,4-DB eq | 187 | 49.74338765 | 1.453030698 | 0.470057227 | 3.204380848 | 1.193904914 | 127.5102642 | 0.003406189 | 0.050343658 | 3.371268635 |
| Drinking Water Toxicity | kg 1,4-DB eq | 139.53 | 53.37677994 | 0.328069929 | 0.128620639 | 1.890451631 | 0.779019919 | 80.88399542 | 0.00415486 | 0.006443846 | 2.133419556 |
| Aquatic Toxicity | kg 1,4-DB eq | 323,768.1 | 118,893.8682 | 979.1907824 | 377.6016196 | 4696.886091 | 2094.217923 | 191,555.8992 | 8.620911215 | 23.01170712 | 5138.825946 |
| Geo Toxicity | kg 1,4-DB eq | 0.63 | 0.453128778 | 0.005408984 | 0.002024162 | 0.019011673 | 0.007078941 | 0.125223336 | $4.09601 \times -10^{-5}$ | $6.17809 \times -10^{-5}$ | 0.022284145 |
| Photochemical Oxidation | kg C2H4 eq | 0.12 | 0.058305601 | 0.00065506 | 0.000257923 | 0.001066332 | 0.00044773 | 0.055914643 | $6.66295 \times -10^{-5}$ | $5.81512 \times 10^{-5}$ | 0.000891232 |
| Acidification | kg SO2 eq | 1.65 | 0.994880014 | 0.018833031 | 0.007158746 | 0.023669518 | 0.012211041 | 0.576482168 | $1.91035 \times -10^{-5}$ | 0.000453662 | 0.013919551 |
| Nutritionism | kg PO4—eq | 0.94 | 0.411389207 | 0.004501924 | 0.001723444 | 0.008838056 | 0.004245764 | 0.506572242 | $1.8924 \times 10^{-5}$ | $5.87169 \times 10^{-5}$ | 0.00658749 |

### 3.3. Comparison of Geopolymer Concrete System and Ordinary Concrete with CML Method

Figure 5 displays the data obtained from the normalization phase of the life cycle assessment of the effects using the CML approach. Normalization has been performed in this part in order to compare the effects of various groups and to standardize the units of the researched effects.

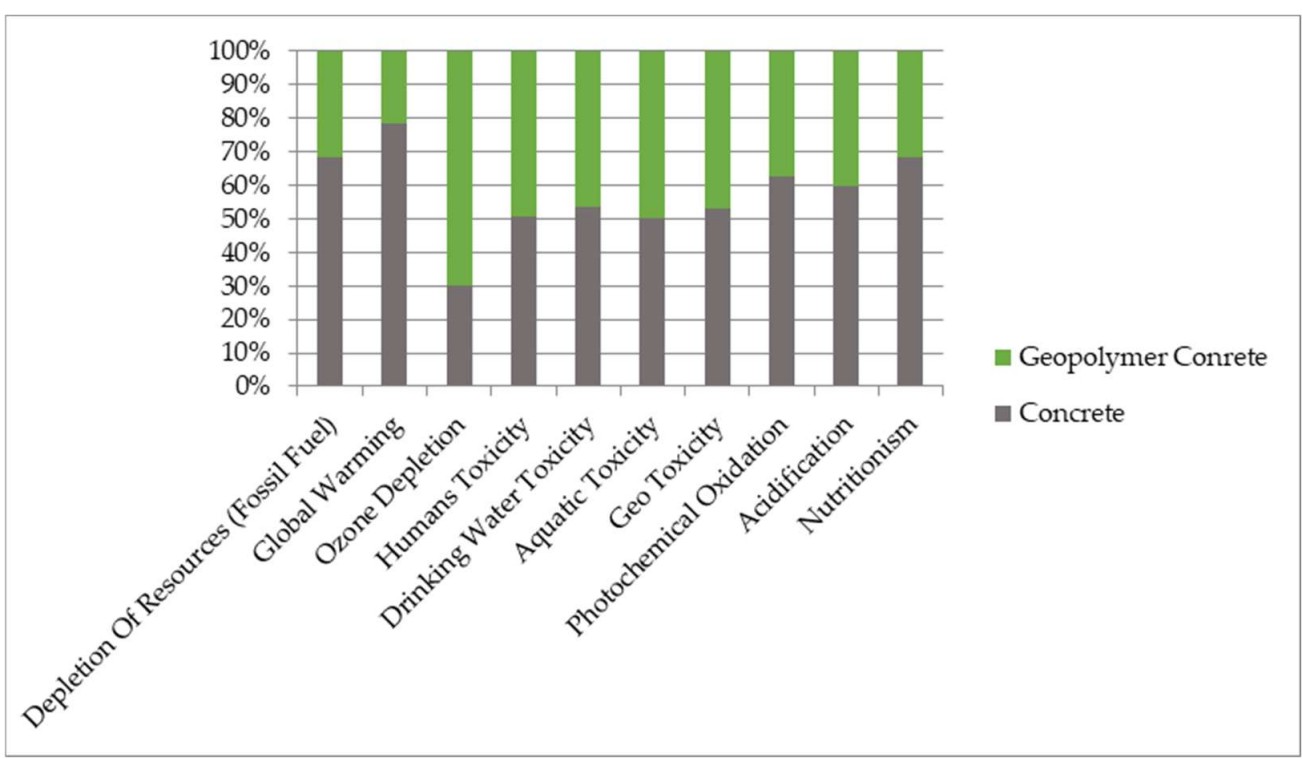

**Figure 5.** The comparison chart of the results of the life cycle evaluation of the production systems of one cubic meter of geopolymer concrete and ordinary concrete, in the stage of normalizing the effects using the CML method.

Comparing the life cycle evaluations of two geopolymer concrete systems based on slag and ordinary concrete using the CML technique revealed that the conventional concrete system is more effective and participatory than the geopolymer concrete system in the majority of environmental degradation classes.

The conventional concrete system has the highest participation in the environmental degradation classes of global warming (greenhouse gas emissions) and resource depletion (fossil fuel) with 81% and 72.45%, respectively, and the lowest participation in the class of ozone depletion with 34%. Previous research shows that the greenhouse gas emissions due to the use of limestone and shale are related to Portland cement concrete compared to geopolymer concrete. In fact, the cement product is responsible for the emission of more than 6% of greenhouse gases [48]. The thermal process of drying clinker is because ordinary concrete uses more fossil fuel. In this regard, Portland cement concrete uses fossil fuel, which is the reason for clinker gridding and clinker drying [49].

In contrast, the geopolymer concrete system contributes the most to the ozone depletion impact category (66 percent) and the least to the global warming impact category (greenhouse gases) (19 percent). For example, Salas et al. [44] examined the geopolymer concrete and conventional concrete LCA based on (NaOH) activator at laboratory macro scale. Their results show that geopolymer concrete illustrates a 64% lower global warming potential than conventional concrete. In other effect categories, the typical concrete system's environmental degradation shows a higher contribution (Figure 5).

According to the results of the study by Bajpai, et al., the authors analyzed the difference in the obtained values for the comparison between conventional and geopolymer concrete systems. They understood that the values of the environmental degradation impact groups of geopolymer and conventional concrete systems are greatly influenced by the mixing design and type of materials used [19,50].

### 3.4. Geopolymer Concrete System by CED Method

The CED technique includes seven categories of resources: two non-renewable energy resources (fossil and nuclear) and five renewable energy resources (biomass, water, wind, sun and geothermal). Table 5 displays the results of a CED-based life cycle evaluation of a slag-based geopolymer concrete system. CED demonstrates the cumulative energy demand of a product based on its production chain (production, transportation and energy) [50,51].

As is shown in Table 5, the total energy required to manufacture one cubic meter of geopolymer concrete containing slag is 2,368,709 MJ. The amount of fossil, nuclear and renewable energy (water, wind, geothermal and solar) required to manufacture one cubic meter of geopolymer concrete is 19,401 MJ (89.9%), 194 MJ (8.8%) and 264.19 MJ (12.4%), respectively (Figure 6).The most consumed sub-sections in the geopolymer concrete production process are sodium silicate sub-sections with a value of 820.0842 MJ, sodium hydroxide sub-sections with a value of 780.506 MJ and aluminosilicate source of smelting furnace slag sub-sections with a value of 365.66 MJ. The least consumed sub-sections are water and electricity, with values of 0.0348 and 8.097 MJ [50,51] (Table 5).

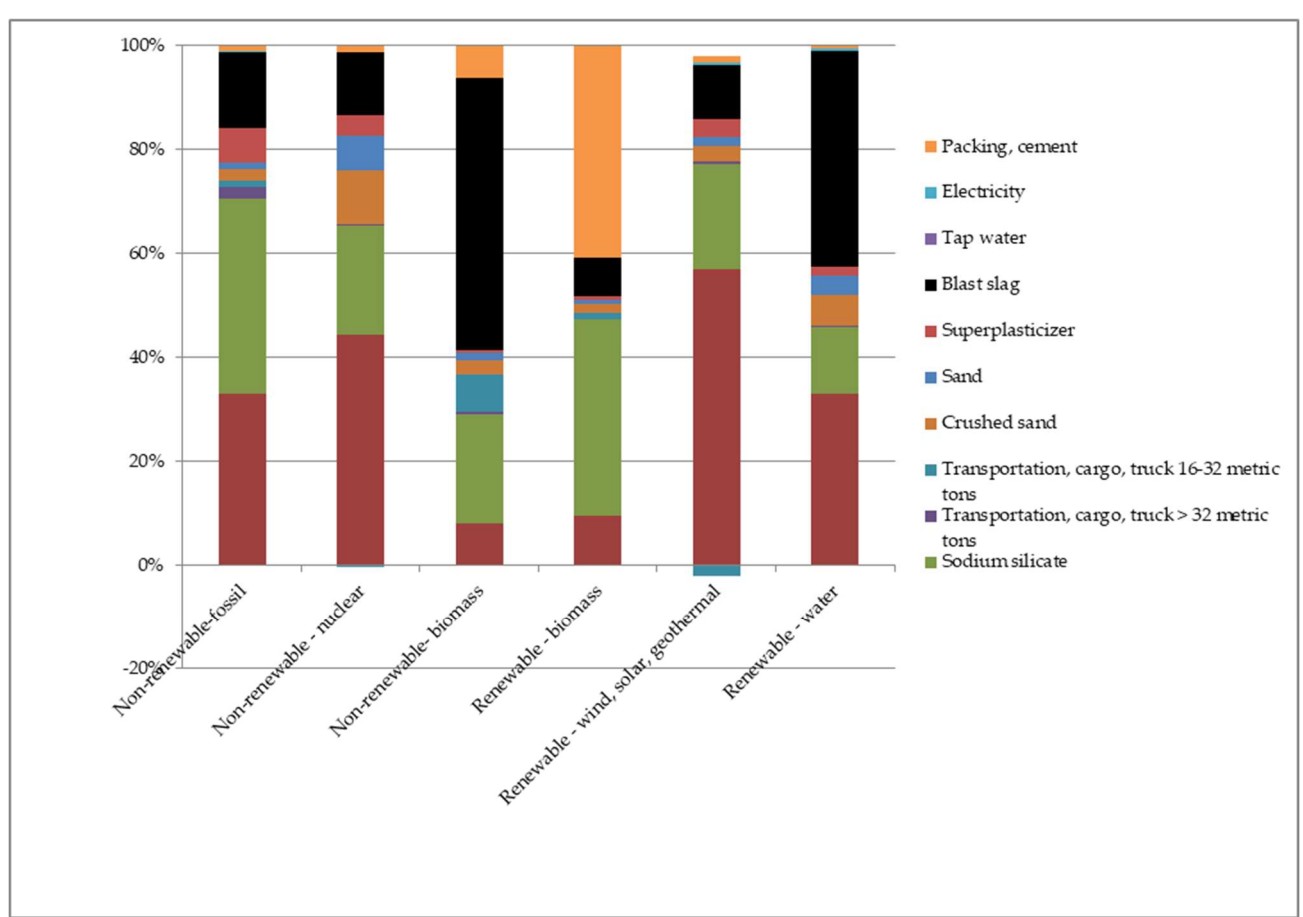

**Figure 6.** Chart of energy demand of fossil fuel, nuclear and renewable energy (water, wind, geothermal and solar) to produce one cubic meter of geopolymer concrete by CED method.

### 3.5. Conventional Concrete System by CED Method

According to the results of a life cycle assessment of conventional concrete using the CED cumulative demand technique, the total amount of energy required to produce one cubic meter of conventional concrete is 6229.38 MJ. One cubic meter of conventional concrete requires 4939.65 MJ (80 percent), 805 MJ (13 percent) and 484.06 MJ (7.768 percent) of fossil, nuclear and renewable energy, respectively (Figure 7).

The conventional concrete production process sections that consume the most energy are the cement sub-sections with a value of 2450.84 MJ, respectively. The sections that consume the least energy are the water sub-sections with a value of 0.348 MJ and the electricity sub-sections with a value of 6.38 MJ (Table 6).

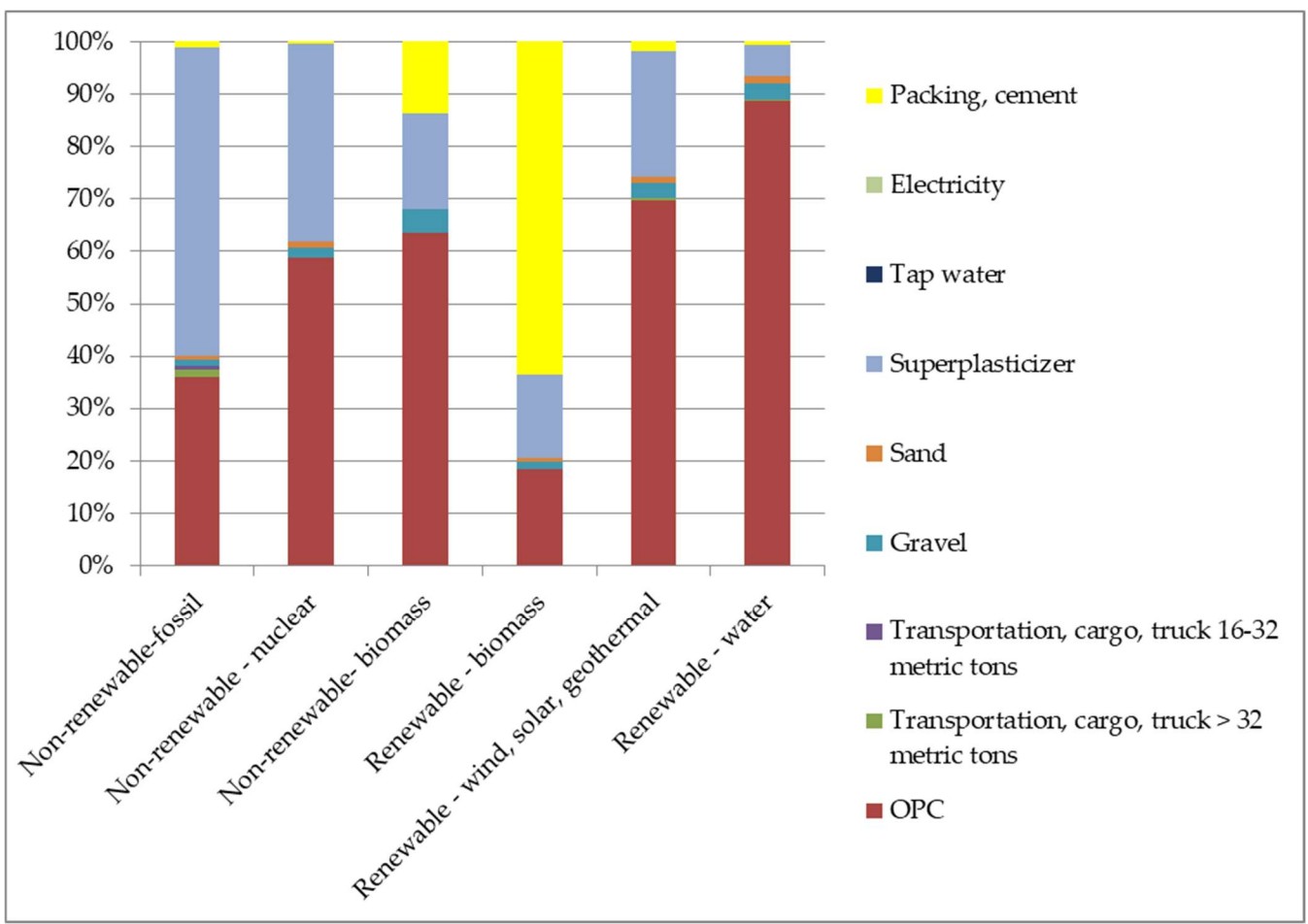

**Figure 7.** Energy demand chart of fossil fuel, nuclear and renewable energy (water, wind, geothermal and solar) to produce one cubic meter of concrete made of ordinary Portland cement using CED method.

**Table 5.** The total energy required to produce one cubic meter of slag-based geopolymer concrete using the CED method.

| | Scale | Total | Sodium Hydroxide | Sodium Silicate | Transportation, Cargo, Truck >32 Metric Tons | Transportation, Cargo, Truck 16–32 Metric Tons | Crushed Sand | Sand | Superplasticizer | Blast Slag | Tap Water | Electricity, High Voltage | Packing, Cement |
|---|---|---|---|---|---|---|---|---|---|---|---|---|---|
| Non-renewable-fossil | MJ | 1912.08 | 631.48 | 715.48 | 45.45 | 24.61 | 39.2 | 26.77 | 125.74 | 277.3 | - | 7.22 | 18.77 |
| Non-renewable-nuclear | MJ | 194.01 | 85.89 | 40.99 | 0.64 | −0.03 | 19.79 | 13.01 | 7.81 | 23.57 | 0.02 | 0.03 | 2.28 |
| Non-renewable-biomass | MJ | 0.21 | 0.02 | 0.04 | - | 0.02 | 0.01 | - | - | 0.11 | - | - | 0.01 |
| Renewable-biomass | MJ | 117.39 | 11.19 | 44.30 | 0.11 | 1.54 | 1.86 | 1.02 | 0.88 | 8.69 | - | 0.01 | 47.77 |
| Renewable-wind, solar, geothermal | MJ | 14.45 | 8.59 | 3.04 | 0.05 | −0.31 | 0.45 | 0.28 | 0.52 | 1.56 | - | 0.08 | 0.17 |
| Renewable-water | MJ | 132.35 | 43.85 | 17 | 0.27 | 0.02 | 7.65 | 4.96 | 2.45 | 54.7 | 0.01 | 0.77 | 0.7 |

**Table 6.** The total energy required to produce one cubic meter of concrete made from ordinary Portland cement using the CED method.

| | Scale | Total | OPC | Transportation, Cargo, Truck >32 Metric Tons | Transportation, Cargo, Truck 16–32 Metric Tons | Gravel | Sand | Superplasticizer | Tap Water | Electricity, High Voltage | Packing, Cement |
|---|---|---|---|---|---|---|---|---|---|---|---|
| Non-renewable-fossil | MJ | 4939.65 | 1415.47 | 61 | 22.82 | 47.32 | 25.51 | 2321.24 | 0.05 | 0.05 | 39.89 |
| Non-renewable-nuclear | MJ | 805.45 | 716.7 | 0.097 | 0.36 | 23.89 | 12.4 | 460.06 | 0.21 | 0.21 | 4.8 |
| Non-renewable-biomass | MJ | 0.22 | 0.14 | - | - | 0.01 | - | 0.04 | - | - | 0.03 |
| Renewable-biomass | MJ | 160.01 | 29.45 | 0.16 | 0.08 | 2.24 | 0.97 | 25.47 | 0.01 | 0.01 | 101.55 |
| Renewable-wind, solar, geothermal | MJ | 20.41 | 14.23 | 0.08 | 0.03 | 0.54 | 0.27 | 4.89 | - | - | 0.36 |
| Renewable-water | MJ | 303.64 | 269.65 | 0.37 | 0.13 | 9.24 | 4.72 | 17.95 | 0.08 | 0.08 | 1.5 |

### 3.6. Comparison of Geopolymer Concrete System and Ordinary Concrete by CED Method

The findings of comparing the life cycle assessment of two geopolymer concrete systems based on slag and ordinary concrete by CED technique showed that the cumulative energy computed for the production of one cubic meter of geopolymer concrete is 2367.709 MJ, which is 62% less than cumulative energy in the production of one cubic meter of ordinary concrete (6229.619 MJ). Overall, the geopolymer concrete can decrease energy consumption. For example, Esparham et al. proved that geopolymer concrete can decrease the energy consumption of industry [18]. Even though the main problem of cement, according to the CED method, is the waste of energy in factory and emission fuels, geopolymer concrete has shown to have no problem in this sector [52].

In the research of Daniel Salas et al., making one cubic meter of 5, 5.5 and 15 MPa zeolite-based geopolymer concrete uses 46, 37.3 and 27% less energy than producing one cubic meter of regular 15 MPa concrete [53,54].

This variation in the results may be attributable to the type of binder employed to achieve the 28 day resistance [18,55]. Furthermore, in the study by Esparham et al., in a similar resistance, the amount of cumulative energy produced by one cubic meter of geopolymer concrete was 35% less compared to concrete made with Portland cement. This difference in the results of the article can be due to the type of binder material used in the mixing plan in order to achieve 28 days of higher resistance [45,54,55].

## 4. Conclusions

The most essential goals of sustainable design and sustainable urban development are energy efficiency and limiting the negative environmental impacts of buildings during the life cycle of a building or structure. In contrast, the construction industry consumes more raw resources than other industrial sectors. In terms of total energy consumption, the built environment also contributes the most to greenhouse gas emissions. The global trajectory of cement consumption in the long term is relatively unknown for both CED and CML methods. Major waste streams are generated by construction and demolition activities, most of which are recyclable. To create more environmentally friendly materials, the materials designer must be familiar with both the environmental drivers of new materials and the environmental consequences of standard building materials.

The CML method proved that conventional concrete is more harmless than geopolymer concrete in many aspects such as global warming, humans toxicity, the consumption of fossil fuels, etc. However, geopolymer concrete is more harmful to the ozone layer due to the same base. The CED method proves that Portland base concrete consumed more energy and fuel than geopolymer due to traditional cement pre-preparation methods, such as clinker grinding and drying.

Geopolymers can be created from a number of raw materials or wastes using a variety of methods to achieve qualities that make them suitable for a wide range of uses.

As a result, according to the CED method, they can dramatically reduce energy consumption and carbon dioxide emissions in the construction industry, providing major environmental benefits. In this research, the environmental performance of one cubic meter of geopolymer concrete production systems based on blast furnace slag and ordinary concrete was evaluated using a life cycle assessment (Portland cement).

This effort in sustainable urban development determined the optimal production technology of environmentally friendly concrete for an introduction in the creation of sustainable materials and use in sustainable design.

According to CML method, the slag-based geopolymeric concrete system had the least participation in the environmental degradation impact categories in all impact classes, except for the ozone depletion impact category, which includes: 28% reduction in resources (fossil fuels), global warming (greenhouse gas emissions) at 18%, ozone depletion of 66%, human toxicity 45%, drinking water toxicity 42%, aquatic toxicity 45%, geo toxicity 42%, photochemical oxidation at 32%, acidification of 36% and over-nutrition of 27% had the least participation in the environmental degradation effect classes.

Considering that the production system of one cubic meter of conventional concrete has a higher contribution in all effect categories (according to the effect of ozone layer destruction with a 34% contribution) compared to the production system of one cubic meter of geopolymer concrete, it can have major adverse effects on the health of aquatic and terrestrial ecosystems and even on human health. In addition, the production system of one cubic meter of ordinary concrete during its lifetime consumes 62% more direct and indirect energy than the system of producing one cubic meter of slag-based geopolymer concrete. In order to further reduce the environmental pollution load of geopolymer concrete in sustainable development, the following are suggested:

1. The method of sodium silicate synthesis from agricultural waste should be used in order to reduce energy consumption and environmental pollution load of geopolymer concrete;
2. The combination of different aluminosilicate materials to reduce the consumption of sodium silicate in the mixing plan;
3. Using geopolymer cement as a suitable alternative to Portland cement in the construction industry.

In general, there is an urgent need to know the durability and lifetime performance of geopolymer systems compared to traditional systems. More studies are needed to improve the technology and strengthen the potential of geopolymer systems in commercial applications in order to reduce environmental impacts.

**Author Contributions:** Conceptualization, A.E., N.I.V., M.H. and M.K.; Data curation, A.E. and M.H.; Formal analysis, A.E. and M.H.; Investigation, A.E. and M.H.; Methodology, A.E. and M.H.; Project administration, A.E., N.I.V. and M.K. Resources, A.E., N.I.V. and M.K. Software, A.E. and M.H.; Supervision, A.E. and N.I.V.; Validation, A.E., N.I.V. and M.K. Visualization, A.E. and M.H.; Writing—original draft, A.E. and M.H.; Writing—review and editing, A.E., N.I.V. and M.K. All authors have read and agreed to the published version of the manuscript.

**Funding:** The research was funded by the Ministry of Science and Higher Education of the Russian Federation as the grant Self-Healing Construction Materials (contract No. 075-15-2021-590 dated 4 June 2021).

**Institutional Review Board Statement:** Not applicable.

**Informed Consent Statement:** Not applicable.

**Data Availability Statement:** The data used to support the findings of this study are included within the article.

**Conflicts of Interest:** The authors declare no conflict of interest. The funders had no role in the design of the study; in the collection, analyses, or interpretation of data; in the writing of the manuscript, or in the decision to publish the results.

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
