# Peer review of "A Study of Modern Eco-Friendly Composite (Geopolymer) Based on Blast Furnace Slag Compared to Conventional Concrete Using the Life Cycle Assessment Approach"

_infrastructures, doi:10.3390/infrastructures8030058_

Round 1

Reviewer 1 Report

The scope of the study is very well organized. The abstract is suitable for the purpose of the study. In the introduction, basic information, literature review, and the difference between the study from the literature are clearly presented. The methods and results used in the study are explained very well.

The following corrections are kindly recommended:

1-      Classification, Characterization, Normalization, and Weighting given in the material method section should be given as a new subtitle.

2-      In some places in the article, the period mark is used before the square bracket. The period is not used after some square brackets. For example;

Sustainable materials are renewable materials that have a good effect on employment and contribute to economic activities based on economy, environment and energy. [10–12].

Consequently, the adaptability and robustness of urban infrastructure is critical for long- 58 term development [6]

These errors need to be corrected throughout the article.

3-      It is stated in the paper that the shipping distance of raw materials was evaluated for all materials based on expert judgment (Table 3).  In my opinion shipping distances vary from region to region and are not exact. Therefore, I recommend removing this statement and Table 3 from the manuscript.

4-      The color scales used in figures 3, 4, 6, and 7 seem very close to each other. It is difficult to distinguish. Can another method be used to describe these graphs?

Author Response

Thanks to your response, the manuscript has been improved in various aspects. Your comments on the article have been applied. I hope you are satisfied with the correction.

The scope of the study is very well organized. The abstract is suitable for the purpose of the study. In the introduction, basic information, literature review, and the difference between the study from the literature are clearly presented. The methods and results used in the study are explained very well.

The following corrections are kindly recommended:

1-      Classification, Characterization, Normalization, and Weighting given in the material method section should be given as a new subtitle.

The notification has been applied

2-      In some places in the article, the period mark is used before the square bracket. The period is not used after some square brackets. For example;

Sustainable materials are renewable materials that have a good effect on employment and contribute to economic activities based on economy, environment and energy. [10–12].

Consequently, the adaptability and robustness of urban infrastructure is critical for long- 58 term development [6]

These errors need to be corrected throughout the article.

The notification has been applied

3-      It is stated in the paper that the shipping distance of raw materials was evaluated for all materials based on expert judgment (Table 3).  In my opinion shipping distances vary from region to region and are not exact. Therefore, I recommend removing this statement and Table 3 from the manuscript.

 The notification has been applied and table 3 has been removed

4-      The color scales used in figures 3, 4, 6, and 7 seem very close to each other. It is difficult to distinguish. Can another method be used to describe these graphs?

The notification has been applied

Reviewer 2 Report

Please check line 135- end of the sentence- 3. Results- it is not clear- is it reference to the literature ?

Line 213 - Dreyer et al.[34] - is it cited in the sentence?

Table No.4 - plese change the format of the table- it would be better if the first line of the table was connected to the following lines.

Conclusion- very useful study, I suggest to investigate in the future also the influence of such material on the quality of indoor air or possible emisions to users of the building.

Author Response

Thanks to your response, the manuscript has been improved in various aspects. Your comments on the article have been applied. I hope you are satisfied with the correction.

Please check line 135- end of the sentence- 3. Results- it is not clear- is it reference to the literature ?

The notification has been applied

Line 213 - Dreyer et al.[34] - is it cited in the sentence?

The notification has been applied

Table No.4 - plese change the format of the table- it would be better if the first line of the table was connected to the following lines.

The notification has been applied

Conclusion- very useful study, I suggest to investigate in the future also the influence of such material on the quality of indoor air or possible emisions to users of the building.

Thank you for your suggestion

Reviewer 3 Report

In order to verify the green and resource sustainable advantages of geopolymer concrete compared with traditional cement concrete, this paper presented the environmental impact of the production, transportation, resource consumption and later practical application of 1m3 slag base geopolymer concrete and cement concrete in the whole life cycle development through (CML) and CED methods. the practical and comprehensive method of CML in quantifying the assessment of environmental effects (such as global warming potential or greenhouse gas effect) and the CED method for evaluating the life cycle energy resources of conventional concrete and geopolymer. The Life Cycle Assessment (LCA) of a Geopolymer Concrete (GC) was elaborated after scaling up the Life Cycle Inventory (LCI) from theoretical research scale to industrial scale. The most relevant raw materials and processes contributing to its environmental performance were identified.

Some specific comments are as follows:

1.Introduction

1、It is suggested to add a comparison between the performance of geopolymer concrete and cement concrete in the introduction, and why geopolymer concrete can replace traditional cement concrete. The advantages of low carbon and sustainable development of geopolymer concrete are introduced.

2、The title of the article emphasizes the research on the sustainable development of geopolymer concrete, which should be compared from its different raw material. Why the comparison between the geopolymer concrete and cement concrete of blast furnace slag base is selected is not just because slag is commonly used raw material.

2.Material and Method

1For impact assessment analysis, this paper simply introduces CML and CDE methods, and does not explain the application of this method.

3.Results and Discussion

1Although this paper is a quantitative research paper, the data in the paper should be further analyzed and interpreted. For example, in Section 3.1, it is shown that the production of geopolymer concrete has a greater harm to the ozone layer, without explaining what causes it. As we all know, geopolymer concrete and cement concrete are produced in a similar way, and both are alkaline. In Section 3.5, the energy consumption in the production of geopolymer concrete is most in the part of sodium silicate and sodium hydroxide.

2The contents of Section 3.2 and Section 3.3 are the same. Please check the contents carefully.

3Although the paper uses two methods to compare and explain the geopolymer concrete system and the ordinary concrete system, the final summary is too simple.

5.Conclusions

1Pay attention to the chapter problems in the passage. Why the jump from chapter 3 to Chapter 5?

2Do lines #425-436 really fit into the conclusion?

3In this paper, the advantages and disadvantages of using CML and CDE methods to evaluate green sustainable development in the life cycle of geopolymer concrete and cement concrete are not well summarized, but only quantitative data. For example, lines #445-451.

Author Response

Thanks to your response, the manuscript has been improved in various aspects. Your comments on the article have been applied. I hope you are satisfied with the correction.

In order to verify the green and resource sustainable advantages of geopolymer concrete compared with traditional cement concrete, this paper presented the environmental impact of the production, transportation, resource consumption and later practical application of 1m3 slag base geopolymer concrete and cement concrete in the whole life cycle development through (CML) and CED methods. the practical and comprehensive method of CML in quantifying the assessment of environmental effects (such as global warming potential or greenhouse gas effect) and the CED method for evaluating the life cycle energy resources of conventional concrete and geopolymer. The Life Cycle Assessment (LCA) of a Geopolymer Concrete (GC) was elaborated after scaling up the Life Cycle Inventory (LCI) from theoretical research scale to industrial scale. The most relevant raw materials and processes contributing to its environmental performance were identified.

Some specific comments are as follows:

1.Introduction

1、It is suggested to add a comparison between the performance of geopolymer concrete and cement concrete in the introduction, and why geopolymer concrete can replace traditional cement concrete. The advantages of low carbon and sustainable development of geopolymer concrete are introduced.

The notification has been applied and some citation has been added.

2、The title of the article emphasizes the research on the sustainable development of geopolymer concrete, which should be compared from its different raw material. Why the comparison between the geopolymer concrete and cement concrete of blast furnace slag base is selected is not just because slag is commonly used raw material.

The title has been changed

2.Material and Method

1、For impact assessment analysis, this paper simply introduces CML and CDE methods, and does not explain the application of this method.

The notification has been applied

3.Results and Discussion

1、Although this paper is a quantitative research paper, the data in the paper should be further analyzed and interpreted. For example, in Section 3.1, it is shown that the production of geopolymer concrete has a greater harm to the ozone layer, without explaining what causes it. As we all know, geopolymer concrete and cement concrete are produced in a similar way, and both are alkaline. In Section 3.5, the energy consumption in the production of geopolymer concrete is most in the part of sodium silicate and sodium hydroxide.

The notification has been applied, and some explanation has been added.

2、The contents of Section 3.2 and Section 3.3 are the same. Please check the contents carefully.

The notification has been applied

3、Although the paper uses two methods to compare and explain the geopolymer concrete system and the ordinary concrete system, the final summary is too simple.

The notification has been applied

5.Conclusions

1、Pay attention to the chapter problems in the passage. Why the jump from chapter 3 to Chapter 5?

The notification has been applied

2、Do lines #425-436 really fit into the conclusion?

Authors think that it is help to understanding the methodology and the work.

3、In this paper, the advantages and disadvantages of using CML and CDE methods to evaluate green sustainable development in the life cycle of geopolymer concrete and cement concrete are not well summarized, but only quantitative data. For example, lines #445-451.

The notification has been applied

Thank your replay.

Round 2

Reviewer 3 Report

The authors have addressed all my concerns.